# Genetic Susceptibility of *HLA* Alleles to Non-Steroidal Anti-Inflammatory Drug Hypersensitivity in the Taiwanese Population

**DOI:** 10.3390/biomedicines11123273

**Published:** 2023-12-11

**Authors:** Szu-Ling Chang, Chih-Hung Lai, Guan-Cheng Lin, Yi-Ming Chen, Mei-Hsuan Lee, Han-Shui Hsu, I-Chieh Chen

**Affiliations:** 1Department of Anesthesiology, Taichung Veterans General Hospital, Taichung 407219, Taiwan; youthdear@gmail.com; 2Department of Postbaccalaureate Medicine, College of Medicine, National Chung Hsing University, Taichung 402202, Taiwan; vincentvghtpe@gmail.com (C.-H.L.); ymchen1@vghtc.gov.com (Y.-M.C.); 3Institute of Emergency and Critical Care Medicine, School of Medicine, National Yang Ming Chiao Tung University, Taipei 112304, Taiwan; 4School of Medicine, National Yang Ming Chiao Tung University, Taipei 112304, Taiwan; 5Department of Medicine and Cardiovascular Center, Taichung Veterans General Hospital, Taichung 407219, Taiwan; 6Department of Medical Research, Taichung Veterans General Hospital, Taichung 407219, Taiwan; t96994024@gs.ncku.edu.tw; 7Division of Allergy, Immunology and Rheumatology, Department of Medical Research, Taichung Veterans General Hospital, Taichung 407219, Taiwan; 8Institute of Biomedical Science and Rong Hsing Research Center for Translational Medicine, National Chung Hsing University, Taichung 402202, Taiwan; 9Institute of Clinical Medicine, National Yang Ming Chiao Tung University, Taipei 112304, Taiwan; meihlee@nycu.edu.tw; 10Division of Thoracic Surgery, Department of Surgery, Taipei Veterans General Hospital, Taipei 112201, Taiwan

**Keywords:** NSAID hypersensitivity, drug allergy, *HLA* alleles, imputation, genotypes

## Abstract

Background: Human leukocyte antigen (*HLA*) genes are important in many immune processes and contribute to many adverse drug reactions. Whether genetic variations in the *HLA* region are associated with non-steroid anti-inflammatory drug (NSAID) hypersensitivity remains uncertain. Therefore, the aim of our study was to identify *HLA* genetic variations in patients with NSAID hypersensitivity in the Taiwanese population. Methods: This hospital-based, retrospective case-control study enrolled 37,156 participants with NSAID exposure from the Taiwan Precision Medicine Initiative (TPMI), who were all genotyped and imputed to fine map *HLA* typing. Our study assigned 1217 cases to the NSAID allergy group and 12,170 controls to a matched group. Logistic regression analyses were utilized to explore associations between *HLA* alleles and NSAID hypersensitivity. Results: Overall, 13,387 patients were genotyped for eight major *HLA* alleles. Allele frequencies were different between the two groups. In the NSAID allergy group, the genotype frequencies of *HLA-A*02:01*, *HLA-A*34:01*, and *HLA-DQA1*06:01* were found to be markedly elevated compared to the control group, a significance that persisted even after applying the Bonferroni correction. Furthermore, the risk of NSAID allergy demonstrated a significant association with *HLA-A*02:01* (OR = 1.29, *p* < 0.001) and *HLA-A*34:01* (OR = 9.90, *p* = 0.001), in comparison to their respective counterparts. Notably, the genotype frequency of *HLA-B*46:01* exhibited a significant increase in the severe allergy group when compared with the mild allergy group. Conclusions: We identified *HLA* genotypes linked to the onset and severity of NSAID hypersensitivity. Our findings establish a basis for precision prescription in future clinical applications.

## 1. Introduction

Non-steroid anti-inflammatory drugs (NSAIDs) are the most commonly prescribed medicine in the current era. They are used to treat pain, fever, and many inflammatory processes [1]. The main mechanism of action is the inhibition of the enzyme Cyclooxygenase (COX), which results in decreased synthesis of prostaglandin and other prostanoids [2]. In addition to the therapeutic usage of NSAIDs, these drugs are well known to exert multiple adverse effects, including cardiovascular, gastrointestinal, nephrotoxic, hepatotoxic, and hypersensitive reactions [3]. The clinical patterns of NSAID hypersensitivity have diverse symptoms, which can often present a diagnostic challenge [4].

NSAID hypersensitivity, an adverse drug reaction (ADR), is a significant cause of drug-related allergies, posing a potential risk of lethal outcomes [5,6]. Early detection of NSAID hypersensitivity is crucial, particularly in patients with no prior exposure to NSAIDs. Studies have shown that the prevalence of NSAID hypersensitivity in the general population ranges from 0.5 to 3.5% [7,8]. However, this prevalence is substantially higher in certain high-risk groups, with 14.89% of severe asthma patients and 9.69% of nasal polyps patients exhibiting NSAID hypersensitivity [9]. These observations suggest a potential link between NSAID hypersensitivity and human leukocyte antigen (*HLA*) gene expression. To delve deeper into the genetic mechanisms underlying NSAID hypersensitivity phenotypes, we employed high-throughput sequencing and characterized the genetic landscape associated with NSAID hypersensitivity phenotypes. Our results may be useful for developing new treatments and therapies for individuals with NSAID hypersensitivity.

The AC-cAMP (adenylate cyclase-cyclic adenosine monophosphate) and GC-cGMP (guanylate cyclase-cyclic guanosine monophosphate) pathways are pivotal signaling cascades within cells, intricately involved in mediating diverse physiological processes. These pathways are crucial for translating the effects of NSAIDs and are closely tied to the body’s inflammatory response. (1) The AC-cAMP pathway involves adenylate cyclase (AC), which catalyzes the conversion of ATP to cyclic AMP (cAMP). Elevated cAMP levels can modulate various cellular functions, influencing inflammation and immune responses. In the context of NSAID allergy, these pathways are disrupted, contributing to the adverse effects observed. Gaseous mediators such as nitric oxide (NO) and carbon monoxide (CO) further complicate this scenario. Some studies have implicated gaseous mediators, such as nitric oxide (NO) and carbon monoxide (CO), in the pathogenesis of NSAID allergy. NO is a signaling molecule that plays a role in various biological processes, including inflammation and vasodilation [10]. (2) The dysregulation of these pathways and the involvement of gaseous mediators contribute to the manifestation of NSAID hypersensitivity [11].

The *HLA* genes of the major histocompatibility complex (MHC) are important in many immune processes, and they are correlated with many autoimmune diseases and hypersensitivity reactions [12]. Studies have shown that specific *HLA* genes are associated with enhanced risk of drug hypersensitivity reactions. For example, in Asian populations, *HLA-B*1502* was identified as a strong genetic risk factor for carbamazepine-induced Stevens–Johnson syndrome (SJS) and toxic epidermal necrolysis (TEN) [13,14]. Similarly, *HLA-B*5801* was associated with an increased risk of allopurinol-induced SJS and TEN in Han Chinese populations [15]. Numerous *HLA* genotypes have a significant association with adverse drug reactions, and several previous studies reported an association of NSAID hypersensitivity with MHC [16,17]. In addition, *HLA-A*02:06, HLA-A*66:01, HLA-B*44:03,* and *HLA-C*12:03* were reported to be associated with NSAID hypersensitivity in the HLA –ADR web-based database [18]. Furthermore, some *HLA* genes were more frequent in the asthma group [19] and the chronic urticarial group [20]. However, whether genetic variations in the *HLA* region are associated with NSAID hypersensitivity remains largely unknown. The purpose of this study was to utilize a hospital-based database to identify *HLA* genetic predisposition in patients who have experienced adverse drug reactions to NSAIDs in the Taiwanese population.

## 2. Materials and Methods

### 2.1. Study Design

This case-control study included 58,091 Taiwanese and was conducted using data from the Taiwan Precision Medicine Initiative (TPMI). From June 2019 to December 2021, all participants were recruited from Taichung Veterans General Hospital (TCVGH) and their electronic health records were collected. Our study cohort comprised 13,387 patients, whose genetic profiles were connected to medical claims data in TCVGH, including demographic characteristics, procedures, examinations, diagnoses, surgeries, medication prescriptions, inpatient services, and outpatient services. This study, which involved human participants, was approved by the Ethics Committee of Taichung Veterans General Hospital Institutional Review Board (IRB no. SF19153A). Written informed consent was obtained from all participants in accordance with the principles defined in the Declaration of Helsinki.

### 2.2. HLA Allele Typing

Our study cohort consisted of 13,387 patients whose DNA from venous blood (2 mL) was genotyped for 8 major MHC Class I (HLA-A, -B, -C) and Class II (-DPA1, -DPB1, -DQA1, -DQB1, -DRB1) loci using an NXType^™^ Class I NGS HLA typing kit and an AllType^™^ NGS 11-Loci Amplification kit (Thermo Fisher Scientific, Waltham, MA, USA). The sequences were analyzed using Axiom HLA Analysis 1.2 (Thermo Fisher Scientific, Waltham, MA, USA), which utilizes advanced imputation methods to enable accurate HLA typing from SNP genotype data over the extended MHC region.

### 2.3. HLA Imputation for Genotype Data

The R library *HLA* Genotype Imputation with the Attribute Bagging (HIBAG) package [21] was utilized to perform HLA imputation for the *HLA* genes *HLA-DQA1*, *HLA-DQB1*, and *HLA-DRB1*. The imputation model used was specific to individuals of Asian ancestry. The resulting imputed data displayed two-field (4-digit) resolution for *HLA* alleles with allele frequencies (AFs) of at least 5.0%.

### 2.4. Participants

As illustrated in Appendix A, the initial cohort included participants who had available genotyping information for *HLA* allele. We extracted participants over the age of 18 years who were exposed to NSAIDs. Data sources included 37,156 participants exposed to NSAIDs; 1217 participants who had an adverse drug reaction report (ADR) were defined as the NSAID allergy group, and 35,939 participants without ADR were defined as the NSAID non-allergy group. Both groups were matched by age and gender at a ratio of 1:10 (Appendix A).

### 2.5. Covariates

In our study, clinical diagnoses were made based on the International Classification of Diseases, Ninth Revision, Clinical Modification (ICD-9-CM) diagnosis codes, requiring a minimum of two outpatient diagnoses or one inpatient diagnosis between January 2009 and December 2021. We obtained comorbidity information from the electronic health records of TCVGH based on ICD-9 diagnostic codes for asthma (ICD-9-CM code 493), atopic dermatitis (ICD-9-CM code 691.8), allergic rhinitis (ICD-9-CM code 477), and urticaria (ICD-9-CM code 708). Then, we compared the allergy severity and *HLA* allele typing. According to the allergy severity level [22], patients with severe systemic reaction (Grade III) were defined as having severe allergy, and those with local reaction/mild-to-moderate systemic reaction (Grade I/II) were defined as having mild allergy.

### 2.6. Statistical Analysis

SAS version 9.4 software (SAS Institute Inc., Cary, NC, USA) was utilized to perform the data analysis. Participants’ characteristics were presented as mean values with standard deviations and percentages across the groups. The chi-square test was utilized to determine the statistical significance between categorical variables, while univariable and multivariable logistic regression analyses were conducted to estimate the associations between *HLA* genotypes and the risks of NSAID allergy. Odds ratios (ORs) and 95% confidence intervals (95% CIs) of *HLA* alleles were estimated using logistic regression models. The final model included significant covariates; the level of significance was set at a *p* value of less than 0.05.

## 3. Results

### 3.1. Characteristics of Participants

This study enrolled a total of 13,387 patients, consisting of 4860 males and 8527 females, who were genotyped for eight major *HLA* alleles. Table 1 shows the basic characteristics of the participants. The mean age of this study group was 56 ± 16 years, and compared to men, women had a greater prevalence of developing NSAID allergy. A higher IgE value was observed in the NSAID allergy group, compared to the control group. Furthermore, the patients in the NSAID allergy group had a lower creatinine level compared to the controls (*p* < 0.001). Compared to the control group, the NSAID allergy group had significantly higher prevalence rates of comorbidities, such as allergic rhinitis (24.7% vs. 20.6%, *p* < 0.001), asthma (13.3% vs. 8.0%, *p* < 0.001), urticaria (9.5% vs. 7.4%, *p* = 0.01), and atopic dermatitis (5.9% vs. 4.4%, *p* = 0.02).

### 3.2. Allele Frequencies and Association between HLA Alleles and NSAID Allergy

The allele frequencies were calculated by means of direct counting. The genotype frequencies of the *HLA* alleles are shown in Table 2, with only those alleles exhibiting statistical significance presented in the table. Appendix A have been meticulously prepared to encompass a comprehensive list of all identified alleles, offering a more thorough overview of the data. In the NSAID allergy group, the genotype frequencies of *HLA-A*02:01*, *HLA-A*34:01*, and *HLA-DQA1*06:01* were found to be significantly elevated compared to the control group, even after applying the Bonferroni correction. Furthermore, the risk of NSAID allergy was significantly associated with *HLA-A*02:01* (OR = 1.29, *p* < 0.001) and *HLA-A*34:01* (OR = 9.90, *p* = 0.001), compared to their counterparts (Table 2).

### 3.3. Association of HLA Alleles and Comorbidities with the Risk of NSAID Allergy

In order to assess the association between *HLA* alleles and comorbidities, we utilized a multiple logistic regression model and evaluated its statistical significance. As shown in Figure 1, the risk of NSAID allergy was significantly associated with *HLA-A*02:01* (OR = 1.29, 95% CI: 1.12–1.48, *p* < 0.001) and *HLA-DRB1*16:02* (OR = 1.22, 95% CI: 1.01–1.47, *p* = 0.003), compared to their counterparts. Additionally, asthma (OR = 1.67, 95% CI: 1.38–2.02, *p* < 0.001) was an independent risk factor for the development of NSAID allergy.

### 3.4. Distribution and Association of HLA Alleles with the Severity of Allergy

Depending on the severity of the NSAID-induced allergy, we stratified the patients into severe and mild allergy groups, respectively (Table 3). In the severe allergy group, the mean age was 57 ± 17 years, which was significantly higher than in the mild allergy group (*p* = 0.01). However, there were no statistically significant differences observed between the blood tests. A higher prevalence of asthma (23.4% vs. 13.2%, *p* = 0.01) was also observed in the severe allergy group, compared to the mild allergy group. The genotype frequencies of *HLA* alleles and their association with the severity of allergy are shown in Table 4. Higher frequencies of *HLA-B*46:01*, *HLA-DPB1*02:02,* and *HLA-DRB*03:01* were observed in the severe allergy group compared to the mild allergy group. Specifically, the genotype frequency of *HLA-B*46:01* was significantly increased in the severe allergy group after Bonferroni correction when compared with the mild allergy group. Additionally, the risk of allergy severity was linked to specific HLA allele genotypes, including *HLA-B*46:01* (OR = 1.68, 95% CI: 1.04–2.71, *p* = 0.035), and *HLA-DPB1*02:02* (OR = 1.95, 95% CI: 1.11–3.42, *p* = 0.020), when compared to their counterparts. Conversely, *HLA-DPB1*05:01* (OR = 0.62, 95% CI: 0.39–0.99, *p* = 0.045) was associated with a reduced risk of severity in the severe allergy group, although these associations did not reach statistical significance after rigorous Bonferroni correction.

### 3.5. Association of HLA Alleles and Comorbidities with Severity of Allergy

*HLA* alleles and comorbidities both had significant impacts on the risk of severity of NSAID-induced allergy. As shown in Figure 2, *HLA-B*46:01* (OR = 1.76, 95% CI: 1.08–2.89, *p* = 0.02) and *HLA-DPB1*02:02* (OR = 1.86, 95% CI: 1.03–3.34, *p* = 0.04) were significantly associated with an increased risk of allergy severity. Similarly, the risk of severity of allergy was strongly associated with asthma (OR = 2.01, 95% CI: 1.10–3.68, *p* = 0.02) using the multiple logistic regression model (Figure 2).

## 4. Discussion

NSAID hypersensitivity likely involves multiple mechanisms, including genetic predisposition, drug metabolism, pharmacokinetics, and environmental factors. Our novel finding implies that *HLA* allele frequency distribution differed between the NSAID allergic and control groups. Certain potential *HLA* alleles could potentially identify NSAID hypersensitivity in the Taiwanese population. In this study, we demonstrated an association between *HLA-A*02:01*, *HLA-A*34:01,* and NSAID-induced ADR. Patients with *HLA-A*02:01* or *HLA-A*34:01* had significantly increased NSAID allergic risk regardless of comorbidities of asthma. With respect to the severity of NSAID allergy, *HLA-B*46:01* and *HLA-DPB1*02:02* had a higher risk; in contrast, *HLA-DPB1*05:01* had a decreased risk of allergy severity. In terms of the comorbidities of asthma, *HLA-B*46:01* and *HLA-DPB1*02:02* had a higher risk, although these associations did not reach statistical significance after rigorous Bonferroni correction.

In our study, the reason we chose *HLA* genes as the target for NSAID hypersensitivity is that more allergic diseases, such as allergic rhinitis, atopic dermatitis, and asthma, are found in patients with NSAID allergy [23,24], as shown in our data. In the current study, asthma had a stronger correlation (OR = 1.67, 95% CI: 1.38–2.02, *p* < 0.001) with the development of NSAID allergy. In previous studies, various *HLA* genes have been demonstrated to be associated with asthma, despite inconsistent findings [25,26,27]. Therefore, the *HLA* gene could be a possible anchor for genetic targeting in this group.

Previous evidence showed correlations between *HLA* genes and drug hypersensitivity reactions [28,29,30], and therefore, pharmacogenetic screening is a possible way to detect patients at high risk of severe drug reactions. As in our study cohort, we found that patients with certain *HLA* alleles tended to have a greater risk of NSAID allergy (*HLA-A*02:01*, *HLA-A *34:01*, *HLA-DQA1*06:01*, *HLA-DRB1*12:02*, *HLA-DRB1*16:02*). In the previous literature, *HLA-DRB* and *HLA-DQ* variability were shown to be associated with aspirin-related hypersensitivity, although the sample size was small [31,32,33,34]. For example, *HLA-DRB* and *HLA-DPB1* showed a correlation with aspirin-induced asthma [33]. *HLA-DRB1*1302*, *HLA-DQB1*0609*, and *HLA-DPB1*0201* were found to be related to aspirin-induced urticaria [32]. In the HLA–ADR web-based database, *HLA-A*02:06* and HLA-B*44:03 had a higher NSAID ADR risk [18]. Our study has a much larger sample size and demonstrated a different pattern of *HLA* allele frequency among NSAID allergy patients; however, the alleles identified in our study were not consistent with those found in the previous studies, potentially due to variations in ethnicity and epigenetic profiles.

In addition to *HLA* polymorphisms, many genetic studies have been shown to have a relationship with NSAID hypersensitivity, such as the *ALOX* 15 gene in the arachidonic acid pathway and DAO in histamine metabolism [35,36,37]. The antigen-presenting cells are the key to activating an allergy reaction, which highlights the importance of *HLA* genes [38,39]. Therefore, it is rational to use them as a genetic marker for hypersensitivity reactions. *HLA* has an important role in antigen-presenting cells to distinguish self from non-self peptides and trigger naïve T cell immunity, and therefore, it is the initial step for many drug hypersensitivity reactions [29]. The different *HLA* allele variations could explain the diversity of ADR. Many studies have discussed the *HLA*–ADR relationship that has been identified in web-based databases. However, there are few data on NSAID hypersensitivity [18].

Genotyping *HLAs* can be useful for identifying individuals who are at higher risk of developing severe adverse reactions to certain drugs, allowing healthcare providers to avoid prescribing these drugs to these individuals, thus reducing the risk of adverse reactions. In the current study, we had a large cohort which contained 37,156 patients with NSAID exposure. We found different *HLA* allele polymorphisms between the NSAID hypersensitivity and control groups.

Nevertheless, this study had certain limitations. First, it is worth noting that our study is retrospective in nature, and as such, the diagnosis of NSAID hypersensitivity was not confirmed through provocation tests. We did not differentiate between immediate hypersensitivity, delayed hypersensitivity, or pseudo-allergic reactions. Additionally, the potential existence of cross-reactivity among different classes of NSAIDs was not explored in this study. To gain a more comprehensive understanding, future research should consider prospective study designs with a focus on confirming our findings through detailed diagnostic assessments. Second, our study did not delve into the realm of epigenetic factors. The influence of epigenetic modifications on gene expression, mediated by various environmental factors and experiences, could contribute to phenotypic variations within the Taiwanese population. Further research is needed to unravel how these epigenetic changes may impact gene expression and its role in NSAID hypersensitivity. Third, it is important to recognize that our analysis was limited to genetic variation within the *HLA* region. While this is a significant factor in hypersensitivity reactions, there may be other genetic pathways outside the *HLA* region that play a role. Future prospective studies should aim to integrate multiple genetic pathways and investigate their interplay to provide a more comprehensive understanding of NSAID hypersensitivity.

## 5. Conclusions

In this retrospective case-control study, our findings underscore the clinical significance of certain *HLA* alleles in the context of NSAID hypersensitivity. Specifically, we identified a significant association between *HLA-A*02:01* and *HLA-A*34:01* with NSAID hypersensitivity, shedding light on the genetic factors involved in this condition. Furthermore, our study revealed that individuals carrying *HLA-B*46:01* and *HLA-DPB1*02:02* alleles faced a significantly elevated risk of severe NSAID allergy. These results not only enhance our understanding of the genetic basis of NSAID hypersensitivity but also have clinical implications for risk assessment and management. The recent identification of *HLA* alleles associated with NSAID allergy severity highlights personalized medicine potential. Tailoring treatments based on individual *HLA* status holds promise for safer interventions. Future studies should integrate clinical phenotypes to grasp genetics, drug metabolism, and environmental factors’ interplay in NSAID hypersensitivity, advancing precision medicine, and understanding drug reactions in practice.

## Figures and Tables

**Figure 1 biomedicines-11-03273-f001:**
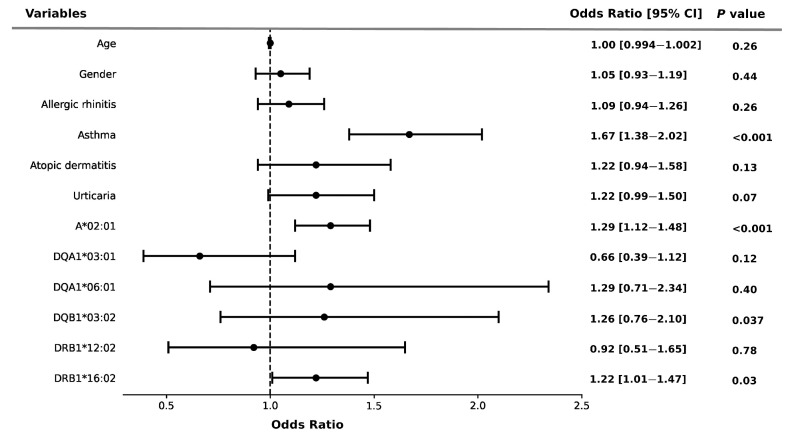
Risk of NSAID allergy and *HLA* alleles in the participants. Error bars represent the 95% confidence intervals of the odds ratios. Multivariate logistic regression adjusted by age, gender, and potential confounders.

**Figure 2 biomedicines-11-03273-f002:**
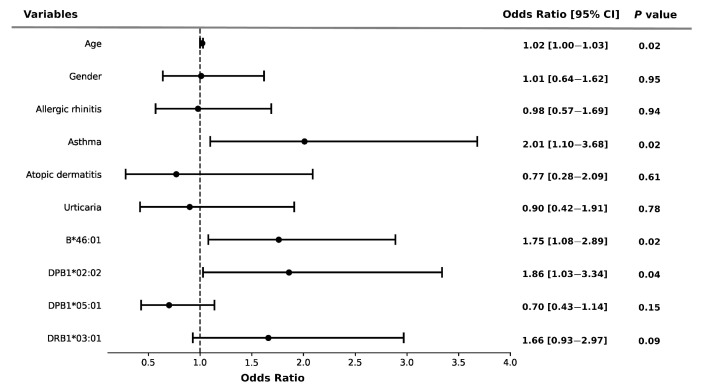
Risk of NSAID allergy severity and *HLA* alleles in the participants. Error bars represent the 95% confidence intervals of the odds ratios. Multivariate logistic regression adjusted by age, gender, and potential confounders.

**Table 1 biomedicines-11-03273-t001:** Baseline characteristics of the participants.

Variables	Total *n* = 13,387		NSAID Allergy *n* = 1217		Control *n* = 12,170		*p* Value
	*n*	(%)	*n*	(%)	*n*	(%)	
Age, years ^a^	56 ± 16		55 ± 16		56 ± 16		0.46
Sex (n, %) ^b^							
female	8527	(63.7)	772	(63.4)	7755	(63.7)	
male	4860	(36.3)	445	(36.6)	4415	(36.3)	0.87
BMI ^a^	25.1 ± 4.7		24.9 ± 4.6		25.1 ± 4.7		0.34
WBC ^a^	7317 ± 2278		7228 ± 2176		7326 ± 2288		0.16
Hemoglobin (HgB) ^a^	12.9 ± 1.7		13.1 ± 1.7		12.9 ± 1.8		0.001
PLT ^a^	248 ± 68		252 ± 703,580		248 ± 68		0.06
Creatinine ^a^	1.22 ± 1.54		1.05 ± 1.06		1.23 ± 1.58		<0.001
GOT ^a^	28 ± 20		28 ± 20		28 ± 20		0.93
GPT ^a^	28 ± 20		28 ± 19		28 ± 20		0.52
IgE ^a^	223 ± 641		307 ± 611		212 ± 644		0.02
Eosinophils ^a^	276 ± 3283		235 ± 267		281 ± 3488		0.87
Tryptase ^a^	5.81 ± 5.84		2.27 ± 0.88		7.23 ± 6.41		0.16
Comorbidity (n, %) ^b^							
Allergic rhinitis							<0.001
No	10,583	(79.1)	917	(75.3)	9666	(79.4)	
Yes	2804	(20.9)	300	(24.7)	2504	(20.6)	
Asthma							<0.001
No	12,247	(91.5)	1055	(86.7)	11,192	(92)	
Yes	1140	(8.5)	162	(13.3)	978	(8)	
Atopic dermatitis							0.02
No	12,778	(95.5)	1145	(94.1)	11,633	(95.6)	
Yes	609	(4.5)	72	(5.9)	537	(4.4)	
Urticaria							0.01
No	12,375	(92.4)	1101	(90.5)	11,274	(92.6)	
Yes	1012	(7.6)	116	(9.5)	896	(7.4)	
Autoimmune							0.65
No	10,219	(76.3)	936	(76.9)	9283	(76.3)	
Yes	3168	(23.7)	281	(23.1)	2887	(23.7)	

^a^ Continuous variables were expressed as mean ± standard deviation (SD) and were analyzed using Student’s *t*-test for normal data distributions. ^b^ Categorical variables were expressed as numbers (percent) and were analyzed using the chi-square test. Abbreviations: BMI: body max index; WBC: white blood cell; HgB: hemoglobin; PLT: platelet; GOT: glutamic-oxalacetic transaminase; GPT: glutamyl pyruvic transaminase.

**Table 2 biomedicines-11-03273-t002:** The genotype frequencies of the *HLA* alleles in the participants.

Variables		NSAID Allergy		Control		*p* Value ^a,c^	Risk of NSAID Allergy	
		*n* = 2434 (Alleles)		*n* = 24,340 (Alleles)				
		*n*	(%)	*n*	(%)		OR (95% CI)	*p* Value ^b,c^
A*02:01	No	2146	(88.2)	21,958	(90.2)			
	Yes	288	(11.8)	2382	(9.8)	0.002 *	1.29 (1.12–1.48)	<0.001 *
A*34:01	No	2430	(99.8)	24,336	(99.9)			
	Yes	4	(0.2)	4	(0.0)	<0.001 *	9.90 (2.47–39.7)	0.001 *
DQA1*03:01	No	2286	(93.9)	22,517	(92.5)			
	Yes	148	(6.1)	1823	(7.5)	0.013	0.79 (0.66–0.95)	0.011
DQA1*06:01	No	2174	(89.3)	22,165	(91.1)			
	Yes		(10.7)	2175	(8.9)	0.005 *	1.19 (1.03–1.39)	0.020
DQB1*03:02	No	2268	(93.2)	22,367	(91.9)			
	Yes	166	(6.8)	1973	(8.1)	0.028	0.82 (0.69–0.98)	0.027
DRB1*12:02	No	2161	(88.8)	22,017	(90.5)			
	Yes	273	(11.2)	2323	(9.5)	0.009 *	1.17 (1.01–1.36)	0.032
DRB1*16:02	No	2285	(93.9)	23,101	(94.9)			
	Yes	149	(6.1)	1239	(5.1)	0.032	1.21 (1.01–1.46)	0.038

^a^ Categorical variables were expressed as numbers (percent) and were analyzed using the chi-square test. ^b^ Univariate logistic regression adjusted by age and gender. ^c^ Adjustment for multiple comparisons: Bonferroni test. * *p* value ≦ 0.05.

**Table 3 biomedicines-11-03273-t003:** Characteristics of the study participants with allergy severity levels.

Variables	Total *n* = 746		Severe Allergy *n* = 94		Mild Allergy *n* = 652		*p* Value
	*n*	(%)	*n*	(%)	*n*	(%)	
Age, years ^a^	54 ± 15		57 ± 17		53 ± 15		0.01
Sex (n, %) ^b^							
female	485	(65.0)	60	(63.8)	425	(65.2)	
male	261	(35.0)	34	(36.2)	227	(34.8)	0.89
BMI ^a^	25.0 ± 4.6		24.9 ± 3.8		25.0 ± 4.7		0.84
Allergic rhinitis ^b^							
No	542	(72.7)	66	(70.2)	476	(73.0)	
Yes	204	(27.3)	28	(29.8)	176	(27.0)	0.66
Asthma ^b^							
No	638	(85.5)	72	(76.6)	566	(86.8)	
Yes	108	(14.5)	22	(23.4)	86	(13.2)	0.01
Atopic dermatitis ^b^							
No	703	(94.2)	89	(94.7)	614	(94.2)	
Yes	43	(5.8)	5	(5.3)	38	(5.8)	1.00
Urticaria ^b^							
No	668	(89.5)	85	(90.4)	583	(89.4)	
Yes	78	(10.5)	9	(9.6)	69	(10.6)	0.91
Autoimmune ^b^							
No	562	(75.3)	75	(79.8)	487	(74.7)	
Yes	184	(24.7)	19	(20.2)	165	(25.3)	0.35

^a^ Continuous variables were expressed as mean ± standard deviation (SD) and were analyzed using Student’s *t*-test for normal data distributions. ^b^ Categorical variables were expressed as numbers (percent) and were analyzed using the chi-square test.

**Table 4 biomedicines-11-03273-t004:** Genotype frequencies of allergy severity in the participants.

Variables		Severe Allergy		Mild Allergy		*p* Value ^a,c^	Risk of Severe Allergy	
		*n* = 188 (Alleles)		*n* = 1304 (Alleles)				
		*n*	(%)	*n*	(%)		OR (95% CI)	*p* Value ^b,c^
B*46:01	No	154	(81.9)	1154	(88.5)			
	Yes	34	(18.1)	150	(11.5)	0.014 *	1.68 (1.04–2.71)	0.035
DPB1*02:02	No	168	(89.4)	1225	(93.9)			
	Yes	20	(10.6)	79	(6.1)	0.0280	1.95 (1.11–3.42)	0.020
DPB1*05:01	No	112	(59.6)	671	(51.5)			
	Yes	76	(40.4)	633	(48.5)	0.0450	0.62 (0.39–0.99)	0.045
DRB1*03:01	No	168	(89.4)	1220	(93.6)			
	Yes	20	(10.6)	84	(6.4)	0.0500	1.73 (0.99–3.02)	0.054

^a^ Categorical variables were expressed as numbers (percent) and were analyzed using the chi-square test. ^b^ Univariate logistic regression adjusted by age and gender. ^c^ Adjustment for multiple comparisons: Bonferroni test. * *p* value ≦ 0.05.

## Data Availability

The original contributions presented in the study are included in the article and further inquiries can be directed to the corresponding author.

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
