# Peer review of "Genetic Susceptibility of HLA Alleles to Non-Steroidal Anti-Inflammatory Drug Hypersensitivity in the Taiwanese Population"

_biomedicines, 2023, doi:10.3390/biomedicines11123273_

Round 1

Reviewer 1 Report

Comments and Suggestions for Authors

In this paper, the authors have genotyped 1217 patients diagnosed with NSAID allergy between January 2009 and December 2021, and 1270 controls for a retrospective case control analysis of HLA alleles that could be associated to NSAID allergy in the Taiwanese population. Severel risk HLA alleles were identified.

The main caveat of the study, rising serious doubts about the biological and clinical significance of the results is the selection of cases. As the authors recognize no distinction is made between immediate, delayed, cross-intolerance or selective reactions.

Usually NSAID allergy is an umbrella term that covers a plethora of different clinical conditions. NSAID hypersensitivity reactions can be classified in two groups: those induced by nonspecific immunological mechanisms (non-allergic or cross-intolerance (CI) reactions), or by specific immunological mechanisms (allergic or selective reactions (SR)). The pathogenesis of CI is associated with their pharmacological activity (COX-1 inhibition), with symptoms due to an imbalance in the arachidonic acid pathway, independently of their chemical structure. SRs are mediated by specific IgE- or by a T-cell response and can be induced by a single NSAID or a class of chemically related NSAIDs, with patients tolerating chemically unrelated compounds.

From the immunological point of view HLA alleles could be involved only in IgE or T cell mediated reactions in wich a drug-specific  adaptative immune response develops in allegic individuals. An analysis of HLA association would be desirable only in a selection of selective immediate of delayed NSAID allergic reactions

On the other hand, cross-intolerance reactions to NSAID are mostly associated with genetic variants in enzymes and receptors from the arachidonic acid pathway (eg, ALOX5, ALOX5AP, PTGDR, and CYSLTR1). 

Other comments:

I assume that with such a large sample size, Bonferroni correction should be mandatory for assessement of statistical significance.

Author Response

Author’s Response to reviewer’s comments

Reviewer #1

Comments and Suggestions for Authors

In this paper, the authors have genotyped 1217 patients diagnosed with NSAID allergy between January 2009 and December 2021, and 1270 controls for a retrospective case control analysis of HLA alleles that could be associated to NSAID allergy in the Taiwanese population. Severel risk HLA alleles were identified.

The main caveat of the study, rising serious doubts about the biological and clinical significance of the results is the selection of cases. As the authors recognize no distinction is made between immediate, delayed, cross-intolerance or selective reactions.

Usually NSAID allergy is an umbrella term that covers a plethora of different clinical conditions. NSAID hypersensitivity reactions can be classified in two groups: those induced by nonspecific immunological mechanisms (non-allergic or cross-intolerance (CI) reactions), or by specific immunological mechanisms (allergic or selective reactions (SR)). The pathogenesis of CI is associated with their pharmacological activity (COX-1 inhibition), with symptoms due to an imbalance in the arachidonic acid pathway, independently of their chemical structure. SRs are mediated by specific IgE- or by a T-cell response and can be induced by a single NSAID or a class of chemically related NSAIDs, with patients tolerating chemically unrelated compounds.

From the immunological point of view HLA alleles could be involved only in IgE or T cell mediated reactions in wich a drug-specific adaptative immune response develops in allegic individuals. An analysis of HLA association would be desirable only in a selection of selective immediate of delayed NSAID allergic reactions

On the other hand, cross-intolerance reactions to NSAID are mostly associated with genetic variants in enzymes and receptors from the arachidonic acid pathway (eg, ALOX5, ALOX5AP, PTGDR, and CYSLTR1). 

Author’s response

Thank you for the valuable insights provided by the reviewer. The selection of all NSAID allergy cases in our study was driven by the need to ensure an adequate sample size for statistical analysis and initial exploration. However, we acknowledge a significant limitation in our approach, namely the lack of differentiation between immediate, delayed, cross-intolerance, or selective reactions in the cases.

In response to this feedback, our future research direction will involve a more nuanced categorization of different types of allergic reactions to NSAIDs for a more in-depth investigation. This will include distinguishing reactions induced by non-specific immunological mechanisms (cross-intolerance reactions) from those mediated by specific immunological mechanisms (selective reactions). We plan to specifically analyze the association of HLA alleles in selective immediate or delayed NSAID allergic reactions.

Furthermore, we aim to integrate multiple genetic pathways in future studies, investigating their interplay to provide a more comprehensive understanding of NSAID hypersensitivity. Clinical phenotypes will be a focal point in future studies to better comprehend the underlying mechanisms of NSAID hypersensitivity and to elucidate the relationships among genetic predisposition, drug metabolism, and environmental factors. These considerations are expected to contribute to a more thorough and nuanced understanding of NSAID hypersensitivity and its associated mechanisms.

Other comments:

I assume that with such a large sample size, Bonferroni correction should be mandatory for assessement of statistical significance.

Author’s response

Thank you for your valuable comment regarding the statistical methods employed in our study. We value your suggestion to incorporate the Bonferroni correction for multiple comparisons. Given our substantial sample size, integrating such a correction is vital to uphold the overall significance level and mitigate the risk of Type I errors. We have now included the Bonferroni correction in our analyses, providing a more robust evaluation of statistical significance. Please refer to the revised manuscript in Tables 2 and 4 for details (Please see Table 2 and Table 4).

Reviewer 2 Report

Comments and Suggestions for Authors

Thanks for the opportunity to review the manuscript by Szu-Ling Chang and collaborators. They aimed to identify HLA genetic susceptibility in patients who have experienced adverse drug reactions to NSAIDs in the Taiwanese population.

The manuscript is mainly well-written, and the study design is robust, with a considerable sample size. The inclusion criteria are well described. I have some comments for the authors:

The results are really interesting. The HLA-DRB1*16:02 (OR = 1.21, p=0.038) is associated with a higher risk of NSAID allergy but has a decreased risk of severity HLA-DRB1*16:02 (OR = 0.38, p = 0.043). Please comment on this point in the discussion section.

Clearly, Table 2 only shows HLA alleles with significant statistical values in the case-control comparison but does not offer the possibility of seeing the most common alleles between groups.

I suggest including a sentence describing it (only alleles with statistical...) and a supplementary table/material with all the identified alleles.

In the sentence (line 64) "...using high-throughput sequencing, we investigated the genetic mechanisms involved in..." is imprecise; please note that this type of study design is just descriptive. please reformulate according to this. 

Minor comments: 

In the sentence "Two groups were matched by age and gender at 121 a ratio of 1:10 (Supplementary Figure 1)." please use "Both" instead of "Two"

Line 101 change "MHC" to "HLA" and remove it from the parenthesis.

Please define   HgB meaning (in Table 1). Also, include the meaning of all abbreviations employed in the footnote.

Supplementary material for publishing and non-publishing is the same; is it ok?

Comments on the Quality of English Language

Only minor mistakes, typos mainly.

Example: In the sentence "Two groups were matched by age and gender at 121 a ratio of 1:10 (Supplementary Figure 1)." please use "Both" instead of "Two"

Author Response

Author’s Response to reviewer’s comments

Reviewer 2

Comments and Suggestions for Authors

Thanks for the opportunity to review the manuscript by Szu-Ling Chang and collaborators. They aimed to identify HLA genetic susceptibility in patients who have experienced adverse drug reactions to NSAIDs in the Taiwanese population.

The manuscript is mainly well-written, and the study design is robust, with a considerable sample size. The inclusion criteria are well described. I have some comments for the authors:

The results are really interesting. The HLA-DRB1*16:02 (OR = 1.21, p=0.038) is associated with a higher risk of NSAID allergy but has a decreased risk of severity HLA-DRB1*16:02 (OR = 0.38, p = 0.043). Please comment on this point in the discussion section.

Author’s response

We found an interesting result that HLA-DRB1*16:02 (OR = 1.21, p=0.038) is associated with a higher risk of NSAID allergy but has a decreased risk of severity HLA-DRB1*16:02 (OR = 0.38, p = 0.043). We hypothesize that the presence of HLA-DRB1*16:02 alleles in severe allergies is relatively low. To enhance the statistical significance, it may be necessary to expand our patient sample size through additional data collection. This result highlights the complexity of the relationship between HLA alleles and NSAID allergy, emphasizing the need for more extensive research to elucidate the underlying mechanisms and potential clinical implications.

Clearly, Table 2 only shows HLA alleles with significant statistical values in the case-control comparison but does not offer the possibility of seeing the most common alleles between groups.

I suggest including a sentence describing it (only alleles with statistical...) and a supplementary table/material with all the identified alleles.

Author’s response

Thank you for your valuable comment. The table in the manuscript displays only HLA alleles that exhibited statistical significance in the case-control comparison. To address the reviewer's suggestion, we have included a sentence in the manuscript clarifying that only alleles with statistical significance are presented in Table 2. Additionally, we have prepared the supplementary tables containing a comprehensive list of all identified alleles, regardless of statistical significance, to provide a more complete overview of the data (Please see page 5, lines 188-192; Supplementary Table 1 to 8).

In the sentence (line 64) "...using high-throughput sequencing, we investigated the genetic mechanisms involved in..." is imprecise; please note that this type of study design is just descriptive. please reformulate according to this. 

Author’s response

We fully agree the Reviewer 2’s suggestion. The reviewer pointed out that the expression "using high-throughput sequencing, we investigated the genetic mechanisms involved in..." in line 64 is imprecise and suggested that this type of study design is merely descriptive. Therefore, to better reflect the nature of this study, we would rephrase the sentence as follows:

"we employed high-throughput sequencing and characterized the genetic landscape associated with NSAID hypersensitivity phenotypes"

NSAID hypersensitivity, an adverse drug reaction (ADR), is a significant cause of drug-related allergies, posing a potential risk of lethal outcomes [5, 6]. Early detection of NSAID hypersensitivity is crucial, particularly in patients with no prior exposure to NSAIDs. Studies have shown that the prevalence of NSAID hypersensitivity in the general population ranges from 0.5 to 3.5% [7, 8]. However, this prevalence is substantially higher in certain high-risk groups, with 14.89% of severe asthma patients and 9.69% of nasal polyps patients exhibiting NSAID hypersensitivity [9]. These observations suggest a potential link between NSAID hypersensitivity and human leukocyte antigen (HLA) gene expression. To delve deeper into the genetic mechanisms underlying NSAID hypersensitivity phenotypes, we employed high-throughput sequencing and characterized the genetic landscape associated with NSAID hypersensitivity phenotypes (Please see page 2, lines 60-70).

Minor comments: 

In the sentence "Two groups were matched by age and gender at 121 a ratio of 1:10 (Supplementary Figure 1)." please use "Both" instead of "Two"

Author’s response

Thank you for your suggestion. We have replaced “Two” with “Both”. Please see the revised manuscript for details

Line 101 change "MHC" to "HLA" and remove it from the parenthesis.

Author’s response

Thank you for the reviewer's suggestions. I have carefully considered the recommendation to change "MHC" to "HLA" and remove it from the parentheses. However, I would like to further explain why we used "MHC" in the original text.

In our study, we are indeed discussing the major MHC Class I and Class II molecules. However, to precisely denote our research subject, we chose to use the term "MHC." This is because "MHC" is commonly used to broadly refer to the major histocompatibility complex, while "HLA" specifically refers to the human MHC. In this context, we believe that using "MHC" reflects a more comprehensive aspect of our research scope.

Although we understand your suggestion, considering the contextual difference, we ultimately chose to retain "MHC" to maintain a precise description of the research scope. Thank you again for your valuable input, and we will continue to focus on communication and enhance the accuracy of our research.

Please define   HgB meaning (in Table 1). Also, include the meaning of all abbreviations employed in the footnote.

Author’s response

Thank you for your suggestion. We have replaced defined Hemoglobin (HgB) in revised Table 1. Please see the revised manuscript for details (Table 1; page 5, lines 184-185).

Supplementary material for publishing and non-publishing is the same; is it ok?

Author’s response

Thank you for the evaluators' attention. We affirm that our supplementary material has not been previously published. These materials encompass Supplementary Figure 1, an illustrative flowchart detailing the study design, and Supplementary Tables, comprising a comprehensive list of all identified allele genes. This arrangement is intended to ensure the information presented to readers is both exhaustive and comprehensible. Should further discussion or modifications be necessary, we are amenable to adjustments.

Comments on the Quality of English Language

Only minor mistakes, typos mainly.

Example: In the sentence "Two groups were matched by age and gender at 121 a ratio of 1:10 (Supplementary Figure 1)." please use "Both" instead of "Two"

Author’s response

Thank you for your suggestion. We have replaced “Two” with “Both”. Please see the revised manuscript for details (Please see page 3, line 142).

Reviewer 3 Report

Comments and Suggestions for Authors

In their manuscript, the authors analyzed the role of hLA genotypes on hypersensitivity to non-steroidal anti-inflammatory drug therapy. The study was designed and conducted correctly. Results presented clearly, well discussed.
Please elaborate on a few minor aspects:
- were there any patients with angioedema in the group?
- in the introduction, please expand a little on the issue of side effects. In particular, please refer to the AC-cAMP and GC-cGMP pathways. It seems particularly important to refer to the role of gaseous mediators of NO and CO and these processes.
- please expand your conclusions. The wording itself is currently not a conclusion but a summary of the results. Conclusions should indicate the importance of the work in everyday practice

Author Response

Author’s Response to reviewer’s comments

Reviewer 3

Comments and Suggestions for Authors

In their manuscript, the authors analyzed the role of hLA genotypes on hypersensitivity to non-steroidal anti-inflammatory drug therapy. The study was designed and conducted correctly. Results presented clearly, well discussed.
Please elaborate on a few minor aspects:
- were there any patients with angioedema in the group?

Author’s response

Thank you for your comment.

Yes, some of the patients exhibited angioedema as a clinical symptom, and these instances were documented in the Adverse Drug Reaction (ADR) reports.

- in the introduction, please expand a little on the issue of side effects. In particular, please refer to the AC-cAMP and GC-cGMP pathways. It seems particularly important to refer to the role of gaseous mediators of NO and CO and these processes.
Author’s response

Thank you for your suggestion. We have made modifications to the introduction to provide a more in-depth discussion of side effects, with a specific focus on the AC-cAMP and GC-cGMP pathways (Please see page 2, lines 73-87).

Please refers to the revised manuscript.

  • Page 2, lines 73-87

“The AC-cAMP (adenylate cyclase-cyclic adenosine monophosphate) and GC-cGMP (guanylate cyclase-cyclic guanosine monophosphate) pathways are pivotal signaling cascades within cells, intricately involved in mediating diverse physiological processes. These pathways are crucial in translating the effects of NSAIDs and are closely tied to the body's inflammatory response. (1) The AC-cAMP pathway involves adenylate cyclase (AC), which catalyzes the conversion of ATP to cyclic AMP (cAMP). Elevated cAMP levels can modulate various cellular functions, influencing inflammation and immune responses. In the context of NSAID allergy, these pathways are disrupted, contributing to the adverse effects observed. Gaseous mediators such as nitric oxide (NO) and carbon monoxide (CO) further complicate this scenario. Some studies have implicated gaseous mediators, such as nitric oxide (NO) and carbon monoxide (CO), in the pathogenesis of NSAID allergy. NO is a signaling molecule that plays a role in various biological processes, including inflammation and vasodilation [Ref 1]. (2) The dysregulation of these pathways and the involvement of gaseous mediators contribute to the manifestation of NSAID hypersensitivity [Ref 2].”

Reference:

(1) Denninger JW, Marletta MA. Guanylate cyclase and the .NO/cGMP signaling pathway. Biochim Biophys Acta. 1999 May 5;1411(2-3):334-50. doi: 10.1016/s0005-2728(99)00024-9. PMID: 10320667.

(2) Nitric oxide and carbon monoxide in NSAID-induced hypersensitivity reactions. Kim SH, Kim YM. Clin Chim Acta. 2010 Apr;411(4-5):314-20. doi: 10.1016/j.cca.2009.12.016. Epub 2010 Jan 14. PMID: 20080382.

- please expand your conclusions. The wording itself is currently not a conclusion but a summary of the results. Conclusions should indicate the importance of the work in everyday practice

Author’s response

We are grateful for this suggestion and have modified the “Conclusions” section.

Original paragraph:

In this retrospective case-control study, we demonstrated that HLA-A *02:01 and HLA-DRB1 *16:02 were associated with NSAID hypersensitivity. Moreover, HLA-B *15:18, HLA-B *46:01, and HLA-DPB1 *02:02 allele carriers had significantly increased risk of severity of NSAID allergy. Future studies with clinical phenotypes are needed to better understand the underlying mechanisms of NSAID hypersensitivity and to elucidate the relationships among genetic predisposition, drug metabolism, and envi-ronmental factors.

Expanding paragraph:

In this retrospective case-control study, our findings underscore the clinical significance of certain HLA alleles in the context of NSAID hypersensitivity. Specifically, we identified a significant association between HLA-A*02:01 and HLA-DRB1*16:02 with NSAID hypersensitivity, shedding light on the genetic factors involved in this condition. Furthermore, our study revealed that individuals carrying HLA-B15:18, HLA-B46:01, and HLA-DPB1*02:02 alleles faced a significantly elevated risk of severe NSAID allergy. These results not only enhance our understanding of the genetic basis of NSAID hypersensitivity but also have clinical implications for risk assessment and management. Recent identification of HLA alleles associated with NSAID allergy severity highlights personalized medicine potential. Tailoring treatments based on individual HLA status holds promise for safer interventions. Future studies should integrate clinical phenotypes to grasp genetics, drug metabolism, and environmental factors' interplay in NSAID hypersensitivity, advancing precision medicine and understanding drug reactions in practice (Please see page 10, lines 334-346).

Round 2

Reviewer 1 Report

Comments and Suggestions for Authors

The study design has not been improved as no stratification has been performed according to specific clinical entities. Therefore, I think that the relevance of the results is of limited use. As NSAID hypersensitivity is linked to exacerbation of asthma and urticaria, it would be really interesting to see the results when only patients with asthma or urticaria are analyzed.  

To avoid introducing more confusion, I suggest limit Table 2  and Table 3 to those alleles with statistical significance after Bonferroni correction. In addition, results sections in the abstract and in the body of the article should be modified to highlight the relevance of only statistically significant findings.

Minor comments:

I understand that the new paragraph in the introduction has been added in response to an additional reviewer. However, I do not see the relevance of new references 10 and 11 for this study.

Author Response

Author’s Response to reviewer’s comments

Reviewer #1

Comments and Suggestions for Authors

The study design has not been improved as no stratification has been performed according to specific clinical entities. Therefore, I think that the relevance of the results is of limited use. As NSAID hypersensitivity is linked to exacerbation of asthma and urticaria, it would be really interesting to see the results when only patients with asthma or urticaria are analyzed.  

To avoid introducing more confusion, I suggest limit Table 2 and Table 3 to those alleles with statistical significance after Bonferroni correction. In addition, results sections in the abstract and in the body of the article should be modified to highlight the relevance of only statistically significant findings.

Author’s response

Thank you for the insightful feedback on our hospital-based, retrospective case-control study. We appreciate your suggestion regarding the lack of stratification based on specific clinical entities, and we acknowledge that this limitation affects the relevance of our results. Moving forward, we plan to enhance our study design by conducting further analyses that specifically focus on patients with asthma or urticaria, considering their potential exacerbation due to NSAID hypersensitivity. This will allow us to provide a more targeted and clinically relevant perspective.

Regarding your recommendation to limit Tables 2 and 4 to alleles with statistical significance after Bonferroni correction, we appreciate the reviewer's suggestion and have indeed restricted Tables 2 and 4 to alleles demonstrating statistical significance following Bonferroni correction. We have modified the results sections in the abstract and the body of the article to clearly convey the significance of the identified alleles after applying the Bonferroni correction. Please see the revised manuscript for details (Please see page 1, lines 38-45; page 5, lines 193-197; page 7, lines 225-234; Table 2 & Table 4).

Minor comments:

I understand that the new paragraph in the introduction has been added in response to an additional reviewer. However, I do not see the relevance of new references 10 and 11 for this study.

Author’s response

Thank you for your kind reminder. In response to another reviewer's request to include information about the AC-cAMP and GC-cGMP pathways, we added reference 10 (Denninger, J.W. and M.A. Marletta: Guanylate cyclase and the .No/cgmp signaling pathway. Biochim Biophys Acta 1999, 1411:334-350) to provide additional insights into the NO/cGMP signaling pathway. However, we acknowledge that reference 11 may be less suitable for this paragraph. As suggested, we will replace reference 11 with a more relevant citation (Laukeviciene A, Ugincius P, Korotkich I, Lazauskas R, Kevelaitis E. Anaphylaxis of small arteries: putative role of nitric oxide and prostanoids. Medicina (Kaunas). 2010;46(1):38-44. PMID: 20234162).

In future studies, we will strive to more clearly establish the pathway between NO and NSAID hypersensitivity for a more comprehensive understanding of the mechanisms involved.

Reviewer 2 Report

Comments and Suggestions for Authors

Thank you for attending to my previous concerns. A minor mistake should be corrected in the conclusion section: HLA-B15:18, HLA-B46:01; please include the asterisk after the "B", like this: HLA-B*15:18, HLA-B*46:01.

Author Response

Author’s Response to reviewer’s comments

Reviewer #2

Comments and Suggestions for Authors

Thank you for attending to my previous concerns. A minor mistake should be corrected in the conclusion section: HLA-B15:18, HLA-B46:01; please include the asterisk after the "B", like this: HLA-B*15:18, HLA-B*46:01.

Author’s response

Thank you for your detailed revision. We have replaced “HLA-B15:18, HLA-B46:01” with “HLA-B*15:18, HLA-B*46:01”. Please see the revised manuscript for details (Please see page 10, line 333).

Reviewer 3 Report

Comments and Suggestions for Authors

The authors made corrections in accordance with the suggestions. In my opinion, the manuscript may be considered for publication

Author Response

Author’s Response to reviewer’s comments

Reviewer #3

Comments and Suggestions for Authors

The authors made corrections in accordance with the suggestions. In my opinion, the manuscript may be considered for publication

Author’s response

Thank you for your comment.

Round 3

Reviewer 1 Report

Comments and Suggestions for Authors

I understand that clinical data are not available to perform an stratified analysis involving defined clinical entities. Otherwise, the authors would have been able to perform it.

However, I think that the paper has been improved